# Changing Base Without Losing Pace: A GPU-Efficient Alternative to MatMuls in DNNs

## Abstract

Modern AI relies on huge matrix multiplications (MatMuls), whose computation poses a scalability problem for inference and training. We propose an alternative, *GPU native* bilinear operator to MatMuls in neural networks, which offers a three-way tradeoff between: speed, accuracy and parameter count. In particular, this operator requires substantially *fewer* FLOPs to evaluate ($\ll n^3$), yet *increases* the parameter count compared to MatMul ($\gg n^2$). We call this operator *Strassen-Tile* (STL). The key idea behind STL is a local **learnable** change-of-basis, applied on *tiles* of the weight and activation matrices, followed by an *element-wise* product between the tiles, implemented simultaneously via MatMul. The key technical question we study is how to optimize the change-of-basis of a given layer, which is a highly non-convex problem. We show that theory-backed initializations (inspired by fast matrix and polynomial multiplication) lead to substantially better accuracy than random SGD initialization. This phenomenon motivates further algorithmic study of STL optimization in DNNs. Our experiments demonstrate that STL can approximate 4x4 MatMul of **tiles** while reducing FLOPs by a factor of 2.66, and can **improve** Imagenet-1K accuracy of SoTA T2T-ViT-7 (4.3M parameters) while lowering FLOPs. Even with non-CUDA optimized `PyTorch` code, STL achieves wall-clock speedups in the compute-bound regime. These results, together with its theoretical grounds, suggest STL as a promising building block for scalable and cost-efficient AI.

## 1 Introduction

Matrix multiplication (MatMul) is a ubiquitous operation across all fields of science and technology. Specifically, MatMuls are the bottleneck (80%-90% of energy, latency and throughput) of training and inference of deep neural networks (DNNs), both for language and for vision models (Kim et al., 2023; Zhu et al., 2024). Indeed, multiplying large matrices (e.g., 16Kx16K) is considered a prerequisite in any Generative-AI model (Li et al., 2025; Naveed et al., 2024), implying a billion-order FLOP count for merely a million-order IOs. As such, continual increase in computation and energy demands underlying AI breakthroughs poses a real scalability problem.

The reliance on MatMuls is mainly attributed to the emergence of hardware which was optimized for this task (GEMM) – GPUs (and in particular, TensorCores (NVIDIA, 2020; 2023)). This hardware allows for extremely efficient amortization of IO and parallelism of the cubic FLOPs ($\approx n^3$ for $n \times n$ matrices), making it feasible to finetune and run a 10B+ Transformer. Indeed, GPU-optimized training was a pivotal factor in the success of AlexNet (Krizhevsky et al., 2012) and hyper-scaling of DL ever since. This phenomenon, where an algorithmic paradigm prevails because it is most suited to the available hardware and *not necessarily* because it is theoretically superior to alternative ideas, is a widely-believed explanation to the rise of deep learning, that "won the hardware lottery" (Hooker, 2021).

Consequently, most of the (massive) research and engineering efforts targeting inference speedups in DNNs attempt to reduce the complexity of MatMuls without major degradation in model accuracy (see Han et al. (2015); Dadush et al. (2018); Han et al. (2016); Abboud et al. (2023); Frantar and Alistarh (2023a); Li et al. (2022); Xiao and et al. (2023) and references therein). This line of research can be divided into two broad categories, which we discuss next.

The first category is *GPU-friendly compression* techniques, attempting to reduce the multiplication to smaller MatMuls or impose structure on the weight matrices (e.g., low-rank decomposition and

linear sketching (Indyk et al., 2019; Zhang et al., 2017; Hu et al., 2022; Choromanski et al., 2021), channel pruning (He et al., 2017), Tensor products (Panagakis et al., 2021), Structured sparsification (Hu et al., 2024; Wen et al., 2016; Hoefler et al., 2021), FFT-like structured weights (Jagtap et al., 2022; Dao et al., 2020a)). A major drawback of these approaches is that they dramatically reduce the *number of trainable parameters* of the weight matrix, resulting in minor speedups for SoTA models, or a substantial loss of accuracy, even after aggressive finetuning (Huang et al., 2022; Moar et al., 2024).

The second category is using algorithmic techniques for approximate MATMUL, which are **not** GPU-friendly, and require the development of *new hardware*. For example, the use of product-quantization (Stock et al., 2020; Fernández-Marqués et al., 2023), weight-sharing (Desai and Shri-vastava, 2024) or *unstructured* sparsification (Frantar and Alistarh, 2023a; Sun et al., 2023a; Hoefler et al., 2021) indicate that the number of parameters in many industry-scale models can be dramatically reduced with minor accuracy loss (up to $90\%$ sparsity in BERT (Kurtic et al., 2022), but barely above $\sim 50\%$ in SoTA LLMs (Frantar and Alistarh, 2023a)). These techniques require specialized hardware and fail to provide real speedups on TensorCores, which is why they have been re-purposed for model *compression* or CPUs (Li et al., 2023). One exception to this category is weight quantiza-tion (Frantar et al., 2022; Frantar and Alistarh, 2023b; Sun et al., 2023b; Dettmers and et al., 2024; Zhu et al., 2024), which is somewhat *orthogonal* to our work, as it cannot yield asymptotic runtime saving in the dimension, but only of the bit-complexity, which remains $\Omega(n^3)$ for $n \times n$ MATMUL. Moreover, quantization can be done in *conjunction* with the method we present.

The above state of affairs explains why inference acceleration is such a notorious challenge in practice – After all, GPUs are optimized for MATMULs, hence it appears that any *generic* MAT-MUL acceleration technique would simply boil down to multiplying smaller matrices, inevitably decreasing the number of parameters of the network. This raises the following question, which our paper is dedicated to answer:

**Question 1.1.** *Is there a bilinear operator $f(X, W)$ which is both faster than* MATMUL$(X, W)$ *on a GPU,* *and does* not *decrease (even increases) the number of trainable parameters?*

Note this is a purely mathematical question, abstracting away accuracy-loss, which is highly *task-specific*. For reference, the *element-wise* (Hadamard) product of square matrices $X \odot W$ preserves the parameter count, but is not faster than MATMUL on TensorCores (performing $\sim n^2$ IOs for $\sim n^2$ FLOPs has very low computational-intensity (NVIDIA, 2023)).

The only known architecture-independent *GPU-efficient* inference acceleration technique, which doesn't *drastically reduce* the parameter count of the model, is *N:M structured sparsification* (Hu et al., 2024; NVIDIA, 2020). As such, we use 2:4 as a baseline for our approach (quantization can be applied in conjunction with 2:4 as well, which makes it an orthogonal axis of optimization, hence our experiments will not include quantization). Specifically, recent TensorCore generations (succeeding Ampere™) can reduce throughput (both FLOP and IO overhead) by up to a factor of 2, when multiplying two matrices, *one of which* has the following 50% sparsity pattern, henceforth denoted **2:4** . In each 4 memory-consecutive matrix elements, at least two out of the four entries in the block must be zero. Deciding *which* of the two entries in a block of the dense pre-trained weight matrix $W$ to zero out (and how to re-train the remaining non-zeroes) so as to minimize accuracy loss, is a nontrivial optimization problem (Hu et al., 2024; Wen et al., 2016; Hoefler et al., 2021).

Our paper is devoted to answering Question 1.1 in the affirmative. We design a *GPU-native and trainable* bilinear operator, whose evaluation requires $\sim n^3/c$ FLOPs (for arbitrary tunable param-eter $c > 1$, compared to $\sim n^3$ for $n \times n$ naive MATMUL) and $\sim n^2$ IOs, while also preserving (often *increasing*) the number of trainable parameters of the network. Thus, our operator, termed *Strassen-Tile* (STL) is more efficient on GPUs than MATMUL, while potentially improving the net-work's expressivity. We analyze some basic properties of this operator, and show that optimizing the operator is a non-trivial optimization problem.

**Related Work.** Our work bridges two lines of research on fast matrix multiplication: (i) A prac-tical line of work attempting to implement *exact* FMM algorithms (a-la Strassen) for MATMUL on existing hardware (Ahmad et al., 2024; Matsuoka and Kang, 2022; Goto and van de Geijn, 2008); and (ii) A recent theoretical line of work which studies an *approximate* version of Strassen's tensor-decomposition for the MATMUL tensor (Pratt et al., 2025; Alman and Zhang, 2023), obtained by

tweaking the tensor to have faster (asymptotic) runtime with *provable* error guarantees. We combine the approaches to obtain practical algorithms with provable guarantees.

Several recent works (Kovachki et al., 2024; Tschannen et al., 2018; Lee-Thorp et al., 2022; Tolstikhin et al., 2024; Anandkumar et al., 2014; Lee-Thorp et al., 2022; Dao et al., 2020b; Fu et al., 2023) consider tensor-algebra products or other bilinear maps as alternatives for MATMULs, suggesting that they are more "expressive" for learning high-dimensional non-Euclidean data, akin to Kernel methods in ML. While similar in spirit to our work, these techniques still suffer from parameter reduction or require new hardware.

Our work is most directly influenced by the work of Tschannen et al. (2018), who proposed to extend Strassen's Fast-Matrix-Multiplication (FMM) framework (Strassen, 1986) to *learn a* universal ternary ($\{-1, 0, +1\}^{r \times n}$) operator, resulting in a multiplication-free operation (an approach which has gained more interest and success very recently (Zhu et al., 2024)). The key difference in our approach is to apply linear transformations locally on *tiles*, which amortizes the cost of these basis-transformations. This key feature turns this method into a GPU-native operator, and allows us to train *unrestricted* tile-transformations over the Reals (via SGD finetuning), which is computationally infeasible using the "universal" approach of Tschannen et al. (2018).

**Structure of the Paper.** In section 2 we survey the necessary background for STL. In section 3 we present the STL operator and analyze its complexity. In section 4 we present a few experiments that corroborate our analysis. In section 5 we shortly discuss the increase in trainable parameters count, which is discussed in more detail in the supplementary material. In section 6 we discuss the source of STL's initialization points and its theoretical foundations.

## 2 STRASSEN NORMAL FORMS

In his seminal work, Strassen (1986) proved that an operator $f(X, W) : \mathbb{R}^{n \times k} \times \mathbb{R}^{k \times m} \to \mathbb{R}^{n \times m}$ is bilinear if and only if it can be written in the following canonical form, called the *Strassen normal form* (SNF):

$$f(X, W) = \mathbf{D}^\top (\mathbf{E_X} \mathrm{vec}(X) \odot \mathbf{E_W} \mathrm{vec}(W)), \tag{1}$$

where $\mathbf{E_X} \in \mathbb{R}^{r \times nk}$, $\mathbf{E_W} \in \mathbb{R}^{r \times mk}$, $\mathbf{D} \in \mathbb{R}^{r \times mn}$ are universal linear transformations ("$X$-encoding", "$W$-encoding" and "decoding" matrices, respectively), $\mathrm{vec}(X) \in \mathbb{R}^{nk}$ is the *vectorized* matrix $X$ (similarly for $\mathrm{vec}(W)$), and $\odot$ denotes the *element-wise* (Hadamard) multiplication of vectors.

The reason we restrict $f$ to be bilinear, besides capturing a very large class of functions (Strassen, 1986; Panagakis et al., 2021), is that ultimately, we *do* wish to take advantage of GPUs (TensorCores) to compute $f(W, X)$ fast. While the Hadamard product in (1) is a very inefficient GPU operation, we will show in the next section that a *tiled* variation of (1) can be *efficiently* computed on a GPU.

## 3 STRASSEN-TILE OPERATOR STL

Fix some prescribed parameter $r = n^2 c$ for $c > 1$. A natural idea, inspired by Tschannen et al. (2018), is to **learn** a bilinear operator instead of MATMUL through its SNF (1) as part of the layer's parameters. This way $c$ governs the number of FLOPs. We can apply Stochastic Gradient Descent (SGD) to finetune the parameters, by taking gradients with respect to $\mathbf{E_X}, \mathbf{E_W}, \mathbf{D}, W$ (where $W$ is the network's weights matrix). This is possible since a bilinear function is differentiable w.r.t. the encoder / decoder matrices of any SNF (1) presentation of the function.

There are two substantial setbacks for implementing this idea:

1. **Changing base is too expensive**: Suppose $X, W$ are $n \times n$ matrices, then computing the products $\mathbf{E_X} \cdot \mathrm{vec}(X), \mathbf{E_W} \cdot \mathrm{vec}(W) \in \mathbb{R}^{r \times n^2}$ requires $n^2 r \sim n^4 c \gg n^3$ FLOPs. Since the optimization is unrestricted, we cannot assume the matrices have useful structure.

2. **Mat-Vec and Element-wise multiplication are too expensive**: As discussed before, computing the Hadamard product of vectors is highly inefficient on a GPU. Moreover, computing the SNF (1) directly, requires computing a Mat-Vec product with a vector of size $n^2$, which incurs huge IO cost.

A natural way to overcome the aforementioned setbacks is to learn the SNF (1) **on small tiles**. This can be interpreted as a one level divide-and-conquer algorithm. For convenience, we introduce the following notation. For $M \in \mathbb{R}^{n \times m}$, assuming $m, n$ are divisible by $t$, we can view $M$ as an element of $(\mathbb{R}^{t \times t})^{n/t \times m/t}$ via tiling, i.e., we view it as a $n/t \times m/t$ matrix whose elements are from the algebra $\mathbb{R}^{t \times t}$ (*tiles*). We use lower-case letters $i \in [n], j \in [m]$ to denote *scalars* $M_{i,j} \in \mathbb{R}$ and upper-case letters $I \in [n/t], J \in [m/t]$ to denote *tiles* $M_{I,J} \in \mathbb{R}^{t \times t}$.

**Definition 3.1** (STL). *Let $n, k, m \in \mathbb{N}$, $t \in \mathbb{N}$ (tile-size), $\mathbb{N} \ni r > t^2$ (tensor-rank), $(\mathbf{E_X}, \mathbf{E_W}, \mathbf{D}) \in \mathbb{R}^{r \times t^2}$ (encoders / decoder). Assume $t$ divides $n, k, m$. Define $\mathsf{STL} : \mathbb{R}^{n \times k} \times \mathbb{R}^{k \times m} \to \mathbb{R}^{n \times m}$, denoted $\mathsf{STL}(X, W) = X \diamond W$, by setting for $I \in [n/t], J \in [m/t]$:*

$$
\mathsf{vec}(X \diamond W)_{I,J} := \mathbf{D}^\top \left( \sum_{L=1}^{k/t} (\mathbf{E_X} \cdot \mathsf{vec}(X_{I,L})) \odot (\mathbf{E_W} \cdot \mathsf{vec}(W_{L,J})) \right). \tag{2}
$$

Note that the decoding step can be computed **once** for every *output* tile, by linearity. Moreover, we call the sum in the RHS of (2) the *encoding of* $(X \diamond W)_{I,J}$.

**Remark 3.2.** *Since $W$ is constant, we can keep only the encoded tiles in memory, i.e., store $w_{I,J} = \mathbf{E_W} \cdot \mathsf{vec}(W_{I,J}) \in \mathbb{R}^r$. Moreover, we can learn directly on $w_{I,J}$ instead of on $\mathbf{E_W}, W$ separately. As discussed in section 5, this can lead to a parameter increase. We call this the **Fake Encoding** of $W$, since it does not originate from an encoding, but plays the same role.*

### 3.1 FLOPs Complexity Analysis of STL

Let $X \in \mathbb{R}^{n \times k}, W \in \mathbb{R}^{k \times m}$ as before. Let $T(r, t)$ denote the cost (in FLOPs) of a single encoding / decoding matrix-vector multiplication (matrix of size $\mathbb{R}^{r \times t^2}$ and vector of size $t^2$, which is the vectorization of a tile). Since we don't assume any structure on our encoding and decoding matrices, we may assume w.l.o.g that $T(r, t) = \Theta(t^2 r)$. In this notation, the runtime of computing $X \diamond W$ is:

$$
\frac{nk}{t^2} \cdot T(r, t) + \frac{mk}{t^2} \cdot T(r, t) + \frac{mn}{t^2} \cdot T(r, t) + \frac{nkm}{t^3} \cdot r. \tag{3}
$$

In the special case of square $n \times n$ matrices, plugging in $T(r, t) = O(t^2 r)$ into (3) simplifies to $O(r(3n^2 + n^3/t^3))$. One can easily verify that as long as $n > 3t^3$, which will be the case as we will be working with small tiles ($t = 4, 8, 16$), the second term dominates the first. Thus, the *amortized* cost of the encoding and decoding transformations is essentially "free" so long as $n \gg t$. In this case, the overall complexity of $\mathsf{STL}(X, W)$ for square $n \times n$ matrices becomes $O\left(rn^3/t^3\right)$. Hence, the speedup factor over the $O(n^3)$ naive MatMul runtime is approximately $(r/t^3)^{-1} = t/c$. We sum this up in the following corollary:

**Corollary 3.3.** *Assuming $n \gg t$, the FLOP count of $\mathsf{STL}$ for square matrices is $O(rn^3/t^3) = O(n^3 c/t)$.*

### 3.2 GPU-Friendly Implementation of the Element-Wise Product

As mentioned earlier, the element-wise product in (2) has very low GPU utilization. In order to compute (2) efficiently on TensorCores, we suggest the following approach. First, for every $p \in [r]$ we define two matrices – $X^{(p)} \in \mathbb{R}^{n/t \times k/t}, W^{(p)} \in \mathbb{R}^{k/t \times m/t}$ – obtained by extracting the $p$-th entry of all $nk/t^2$ (resp. $km/t^2$) encoded tiles of $X$ (resp. $W$). By abuse of notation, we use upper-case indices for $X^{(p)}, W^{(p)}$, and formally define $X_{I,J}^{(p)} := (\mathbf{E_X} \cdot \mathsf{vec}(X_{I,J}))_p$ (similarly $W_{I,J}^{(p)} := (\mathbf{E_W} \cdot \mathsf{vec}(W_{I,J}))_p$). Second, we similarly define $Y^{(p)}$ to be the extraction of the $p$-th entry of the encoded tiles of $Y := X \diamond W$, i.e., before decoding. Thus, it is given by $Y_{I,J}^{(p)} := \left( \sum_{L=1}^{k/t} (\mathbf{E_X} \cdot \mathsf{vec}(X_{I,L}) \odot (\mathbf{E_W} \cdot \mathsf{vec}(W_{L,J})) \right)_p$. The crucial observation is:

**Claim 3.4.** $Y^{(p)} = X^{(p)} W^{(p)}$, *i.e., it is just a standard* MatMul .

The proof is provided in Appendix A. Thus, computing *all* element-wise products, reduces to $r$ MatMuls. We observe that $\mathsf{vec}((X \diamond W)_{I,J}) = \mathbf{D}^\top \begin{bmatrix} Y_{I,J}^{(1)} & Y_{I,J}^{(2)} & \cdots & Y_{I,J}^{(r)} \end{bmatrix}^\top$.

### 3.3 GPU COMPLEXITY ANALYSIS

Building on the GPU-Friendly implementation of the element-wise product, we present a refined performance analysis on GPUs. For simplicity, we assume $n = k = m$. We assume the input matrix $X$ is given as a 3D-tensor of shape $(n/t, n/t, t^2)$, where $\text{vec}(X_{I,J})$ is indexed by $[I, J, :]$ (we use square brackets for the tensor indexing). Moreover, we assume the weights matrix $W$ is given in **encoded** form as a 3D-tensor of shape $(n/t, n/t, r)$, where $\mathbf{E_W} \cdot \text{vec}(W_{I,J})$ is indexed by $[I, J, :]$. Computing $X \diamond W$ from this starting point can be done in three steps: **(i)** Encode, via MATMUL, the tiles of $X$, obtaining $X^{(p)}$ for every $p \in [r]$; **(ii)** For each $p \in [r]$, compute $X^{(p)} W^{(p)}$ via MATMUL, giving the encoded output $Y^{(p)}$; **(iii)** Decode, via MATMUL, each tile of $X \diamond W$ from $\{Y^{(p)}\}_{p \in [r]}$. `PyTorch` pseudo-code for this algorithm is presented in the supplementary material.

For a matrix $M$ let $|M|$ denote the number of bytes needed to store $M$. We analyze each step, assuming ideal hardware and perfect parallelization:

1. **Step (i)**: The IO cost is $|X| + |\mathbf{E_X}|$ for read, $\sum_{p \in [r]} |X^{(p)}|$ for write. Note that $|\mathbf{E_X}|$ is negligible compared to $|X|$ ($rt^2$ compared to $n^3$), while the latter writing size dominates $\sum_p |X^{(p)}| = (r/t^2) \cdot |X|$. Hence the total IO byte load of this step is $\text{IO}_1 \approx |X| \cdot (1 + r/t^2)$. The total number of FLOPs of this step is $2(n/t)^2 \cdot t^2 \cdot r$, as each of the $(n/t)^2$ tiles of $X$ is mapped to $r$ dimensions by a linear transformation, hence $\text{FLOP}_1 = 2n^2 r$.

2. **Step (ii)**: This step consists of $r$ independent MATMULs, each of squared matrices of size $n/t$. Reading the matrices requires IO $\sum_p(|X^{(p)}| + |W^{(p)}|)$, and writing the output requires IO $\sum_p |Y^{(p)}|$. Since all matrices have the same shape, we conclude the total IO byte load is $\text{IO}_2 \approx 3(\sum_p |X^{(p)}|) = 3 |X| \cdot (r/t^2)$. The total number of FLOPs is $\text{FLOP}_2 = 2r \cdot (n/t)^3$.

3. **Step (iii)**: Same analysis as step (i), with the roles of input and output reversed. Thus, $\text{IO}_3 \approx |X| \cdot (1 + r/t^2)$ and $\text{FLOP}_3 = 2n^2 r$.

Overall, we obtain

$$\text{IO}_{\text{STL}} \approx |X| \cdot (2 + 5r/t^2), \quad \text{FLOP}_{\text{STL}} = 4n^2 r + 2n^3 \cdot (r/t^3).$$

At the same time, naive matrix multiplication of $X$ and $W$ requires $\text{IO}_{\text{naive}} = 3 |X|$, $\text{FLOP}_{\text{naive}} = 2n^3$. Note that $\text{FLOP}_2$ dominates when $n \gg t^3$, which is our regime of interest. Thus the asymptotic speedup in FLOPs is by a factor of $t^3/r = t/c$.

**Example 3.5.** *Let us specialize for $t = 4$, $n = 8192$ and $r = 32$. We get $\text{IO}_{\text{STL}} \approx 12 |X|$ and $\text{IO}_{\text{naive}} = 3 |X|$, which is a 4-fold increase in IO load moving to* STL*. Assuming FP16 calculations, $|X| = 2 \times 8192^2 \approx 1.3 \times 10^8$ (bytes). On the other hand, $\text{FLOP}_{\text{STL}} \approx 5583 \times 10^8$ and $\text{FLOP}_{\text{naive}} \approx 10995 \times 10^8$, suggesting an almost 2-fold speedup.*

In practice, DNNs often chain multiple linear layers, interleaved with non-linear activations. For STL, we could fuse (in the sense of CUDA kernel implementation) step (iii) of the previous layer with step (i) of the current layer, reducing the IO load.

It is difficult to use these estimates to predict the actual speedup that STL can give, because this depends on the hardware kernels that are used for executing the computation, usage of cache and other intricate factors that affect performance. Therefore, we have measured the *actual runtime* required to compute STL matrix multiplication versus vanilla matrix multiplication, for various values of $n$ and $r$ (keeping $t = 4$), and using standard CUDA profiling tools, on H100 architecture with FP16 data type.[1] The interested reader may find a figure describing the resulting estimates in Appendix A.

### 3.4 CONCLUSION

The key properties of the STL operator $X \diamond W$ are summarized as follows:

---

[1] We have done these calculations assuming the fused step of (i) and (iii).

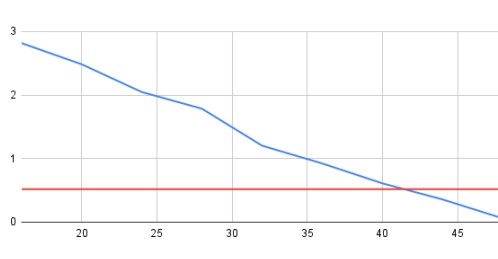 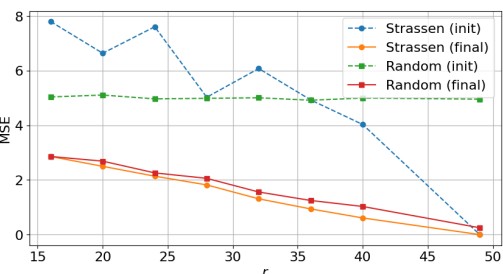

Figure 1: The blue line is estimation of the error of STL, with tile $t = 4$ and tensor rank $r$ from 16 to 48 (for 49, the line is known to cross 0 by Strassen). The red line is our estimate for the corresponding error for $2 : 4$ pruning.

Figure 2: Mean squared error $\downarrow$ of training STL with Strassen based vs. random Gaussian initialization, compared for different values of $r$ before and after training. This shows "smart" initialization maintains an advantage, and is consistent with other methods we have tried.

**Amortization of Encoding**: Each $t \times t$ tile of $X$ and $W$ is encoded only *once*, but used $n/t$ (respectively $m/t$) times in the STL product. As such, the *cost of encoding* is *amortized*, assuming $n \gg t$.

**High GPU utilization**: Assuming $n \gg t^3$, STL can achieve a similar FLOPs per IOs ratio, compared with naive MATMUL.

**Parameter Increase**: This is discussed in section 5. Unlike low-rank, sparse or product-quantization (PQ) approximations, STL does **not** decrease (often increases) the number of trainable parameters of the original linear layer, yet it is *cheaper* than MATMUL$(X, W)$ on GPUs.

## 4 EXPERIMENTS

We perform two classes of experiments. An implementation of STL we've used in our experiments (PyTorch implementation) is available in our public repository.

Class 0 Training encoders and decoders for STL with tile size 4 to approximate $4 \times 4$ matrix multiplication, in the vein of "approximate Strassen Matrix Multiplication". The resulting matrix multiplication residual error is compared against that of 2:4 pruning for the same synthetic random data. We choose 4x4 tiles because it's the simplest scenario for comparing STL and 2:4 , and it suffices because the tiling approach extends the findings to larger matrices.

Class 1 Training from scratch a base *untrained* network, replacing linear layers with STL on tiles of size $t = 4$ and various values of tensor rank $r$. The parameters of the STL encoders and decoders were also trained. For budget and time reasons we worked with vision transformers of the "Token-to-Token" class (Yuan et al., 2021a) with up to $\sim 4.3$M parameters on ImageNet-1K dataset (Deng et al., 2009).

### 4.1 CLASS 0 EXPERIMENTS - SYNTHETIC $4 \times 4$ MATRICES

The first experiment attempts to approximate matrix multiplication of random (Gaussian) $4 \times 4$ single-tile ($t = 4$) matrices $X, W$ using STL for various values of tensor rank $r$, using the Frobenius norm squared of the residual matrix as a loss function, and training on the encoders $\mathbf{E_X}, \mathbf{E_W}$ and decoder $\mathbf{D}$. We used a magnitude based $2 : 4$ pruning strategy on the weight matrices $W$ as benchmark to compare with. As can be seen from Figure 1, we need $r \approx 42$ for STL to match $2 : 4$ pruning. We refer the reader to the supplementary materials for more details.

The result of this experiment is not promising because tensor rank of $r = 42$ for tile size $t = 4$ is unlikely to provide much benefit, if any, from a performance point of view. The result discussed in section 5 explains why we get these rather "disappointing" results, and suggests that when switching the objective function (to real-life AI objectives) and training a network with STL, we can hope to match the 2:4 performance with lower $r$. In fact, as we shall see next, we can even hope to surpass the baseline (linear layer, without STL) with $r$ as low as 24.

For completeness, Figure 2 shows how initialization of the experiment changes the outcomes. In particular, it shows that "smart" initializations are important to achieve good performance, which is evidence to the non-triviality of the optimization problem at hand.

## 4.2 Class 1 Experiments - Training From Scratch with STL

As a model, we experimented with the image classification network T2T-ViT-7 (Yuan et al., 2021b) which has 4.3M parameters (requiring 1.1G FLOPS per 224x224 image). The first step was to repeat the results as reported by Yuan et al. (2021b). We managed to obtain 71.5% accuracy, which is 0.2% lower than claimed there. We attribute this to possible noise stemming from random initialization[2].

Following the baseline result reproduction, we replaced the two MLP linear layers in each of the 7 attention blocks in the network by STL with $r = 16, 24, 32$. For the case $r = 16$ we lost around 2% accuracy compared to base, but for $r = 24, 32$ we improved by close to 0.5% compared to base. Encouraged by this, we replaced not just the MLP linear layers, but also the Q,K,V and the projection linear layers from the attention, thus removing all linear layers from the network trunk, which accounts for 79% of the FLOPS of the entire network[3]. We also did *not* replace activation-×-activation MatMul which we note is easily done with STL but extremely hard to do with $2:4$ sparsification, as it requires on-the-fly sparsification. We used $r = 16, 18, 20, 22, 24, 32, 40, 48, 49$ (the latter case allowing exact matrix multiplication by Strassen) and summarized the results in Table 1 and Figure 3.

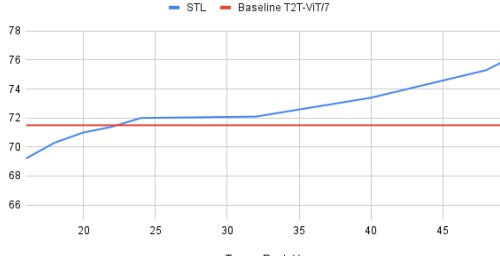

Figure 3: Accuracy ↑ vs. tensor rank $(r)$ for baseline T2T-Vit/7 and for the same architecture with all activation-weight MatMuls in the network trunk replaced by STL with tensor rank $r$. Very similar picture when including the pre-trunk (T2T) network as a followup experiment.

Note that there was no need to adjust any learning parameters. The parameters suggested in the code repository of Yuan et al. (2021b) for T2T-ViT-7 worked out of the box. The results clearly show that, as long as the tensor rank is at least $\approx 24$, we gain accuracy compared to baseline as we increase the tensor rank $r$, and this is likely due to the increase of parameters in the fake encoding parameters (see section 5). For $r < 24$ we lost accuracy points, probably due to loss of expressivity in STL, compared to matrix multiplication, at such a low tensor rank regime.

Further encouraged by these results for $r \geq 24$, we have also replaced all the activation-×-weight linear layers in the T2T part of T2T-ViT-7 with STL, appearing before the network *trunk*. We tested the values $r = 16, 24, 32, 40, 48$. For $r = 16$ we lost an additional 0.7% from this replacement. For $r = 24$ (32) we gained (lost) insignificantly $\leq 0.1\%$, respectively. For $r = 40, 48$ we gained $> 0.4\%$ in each case. This further strengthens our observations about the effect of STL replacement in this regime.

Table 1 summarizes the results in the most ambitious experiment of replacing all linear layers in the body of the network. We provide more technical details about this experiment in the supplementary material.

We are not aware of other work that reports *improvement* on the Imagenet-1k classification problem, when trained from scratch on the Imagenet-1k training split, with a network of similar size, and with weight matrix pruning set at a considerable sparsity rate (in particular structured 2:4 pruning strategies). There are cases where pruning improves accuracy due probably to a regularization effect in the overfitting regime, which is not the case here.

---

[2]Yuan et al. (2021b) mention on the github project page that using 4 GPUs gives slightly lower accuracy than using 8, which may explain the slightly lower baseline we saw when running their code, as we used 4 GPUs.

[3]By trunk we mean the 7 attention blocks. At this point, we did not make replacements in the T2T (Token-to-token) layers preceding the attention blocks, which account for the remaining 21% of the network FLOPS.

### 4.3 CONCLUSION

To summarize, our conclusion from the Class 1 experiment is that in the *under-parameterized* or at most *slightly over-parameterized* case, there is potential of saving factor $> 2.1$ in FLOPs without any loss of accuracy, and in some cases a slight *gain*. However, in the extremely *over-parameterized* case, switching to STL might cause loss of accuracy, due to even higher over-parameterization.

An interesting avenue for future research is to study *larger* ViT architectures for images and/or video, where the dimensions of the matrices justify the use of STL from a performance point of view as well. Another avenue for further experiments is to replace the activation-$\times$-activation MATMULS appearing twice in each attention layer: Once for computation of the so-called *attention matrix*, and again when multiplying the latter with the $V$ (as in Q**K****V**) matrix. We provide more details in the supplementary material.

## 5 PARAMETER INCREASE IN STL

As mentioned before, STL does not only trade off IO and FLOPs, but also the trainable parameter count, which is a measure of the *expressivity* of the network. The parameters $\mathbf{E_X}, \mathbf{E_W}, \mathbf{D}$ offer a negligible addition of parameters to the network. However, as commented before, when training the network with STL, we are free to train directly over $W$ in its encoded form. For every tile of $W$ we have $r \geq t^2$ encoding dimensions, which we refer to as the *Fake Encoding* of $W$. The term comes to emphasize that the vectors cannot be written as the encoding of $W$'s tiles with $\mathbf{E_W}$. A priori, this increases the number of parameters by a factor of $c = r/t^2$.

In the supplementary material, we formally state and prove the following result: Assuming $W$ is a fixed weights matrix, $\mathbf{E_X}, \mathbf{D}$ are also fixed, and $X$ is sampled from a distribution $\mathcal{D}_X$, then optimizing over the *fake encoding* of $W$ to minimize the $L^2$-difference compared to MATMUL, is an optimization problem with the same number of parameters as in $W$. In other words, there is no effective increase in the number of parameters of the network, if we train over the fake encoding instead of $\mathbf{E_W}$.

We make two observations on this result. First, different objective functions (e.g. cross-entropy loss of a network) might prove to have a different effect on the parameter increase, and $L^2$ might be a special case. Second, the result suggests that training STL *after* training the network, i.e., keeping $W$ fixed, might be the problem. Indeed, as we lay out in the next paragraph, our experiments reveal that training a network with STL *from scratch* and optimizing over the fake encoding directly, yields more expressive results.

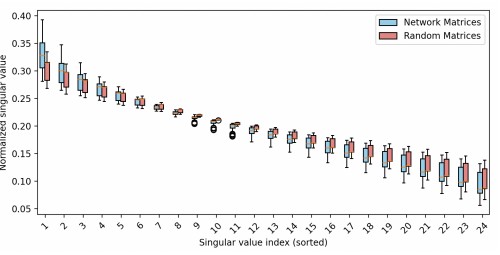

Figure 4: Comparison of singular values between the trained network's (T2T-ViT-7, see subsection 4.2) encoded weights and random matrices of the same sizes. The graph provides evidence that learning with STL occurs in a higher dimensional space.

| Variant | Accuracy ↑ |
|---|---|
| T2T-ViT-7 (Baseline) | 71.5% |
| T2T-ViT-7/STL $r = 16$ everywhere | 69.5% |
| T2T-ViT-7/STL $r = 18$ everywhere | 70.3% |
| T2T-ViT-7/STL $r = 20$ everywhere | 71.0% |
| T2T-ViT-7/STL $r = 22$ everywhere | 71.4% |
| T2T-ViT-7/STL $r = 24$ everywhere | 72% |
| T2T-ViT-7/STL $r = 32$ everywhere | 72.1% |
| T2T-ViT-7/STL $r = 40$ everywhere | 73.4% |
| T2T-ViT-7/STL $r = 48$ everywhere | 75.0% |
| T2T-ViT-7/STL $r = 49$ everywhere | 75.8% |

Table 1: Results for the T2T-ViT/7 model with and without STL replacements. Notice how at the extreme $r = 49$, which can recover exact matrix multiplication, we gain accuracy, most likely due to the increased expressivity from the increased parameter count.

In the class 1 experiment (subsection 4.2), we train a vision transformer from scratch, using STL with tile size $t = 4$ and $r = 24$, *directly training* over the fake encoding. Each 4x4 tile of a weights matrix $W$ corresponds to a fake encoding vector of size $24$. Stacking the vectors side by side we obtain a wide matrix $\mathcal{W}$ with 24 rows. Initialization of $\mathcal{W}$ is by encoding a random Gaussian

matrix $W$ using the matrix $\mathbf{E_W}$ learned in the class 0 experiment (subsection 4.1). The rank of $\mathcal{W}$ before training is at most 16, since the encoded blocks $\mathbf{E_W} \cdot \text{vec}(W_{I,J})$ all belong to the same 16-dimensional sub-space. However, after training, we compute the spectrum (singular values) of $\mathcal{W}$ and observe it uses all 24 possible directions. The results are described in Figure 4. We can conclude that the training process indeed escapes the low dimensional space, showing the fake encoding are utilized.

The key takeaway is that trying to approximate a **trained** linear layer using MATMUL, in the $L^2$ sense, is not the correct approach with STL. Instead, training the network with STL from the ground up, directly on the fake encoding space, increases the number of parameters and possibly increases accuracy.

## 6 THEORETICAL FOUNDATIONS AND INITIALIZATION

The SNF (1) may seem unnatural at first glance, as it interprets a bilinear function $f$ as a change of basis into an $r$-dimensional space. However, for a specific family of bilinear operators, this has a very clean interpretation. Convolution operators of abelian (commutative) groups, can be written, by the *convolution theorem* (Cooley and Tukey, 1965), as follows: Let $\mathbf{F}$ denote the Fourier transform matrix for the underlying group, let $\widehat{\mathbf{E_X}}, \widehat{\mathbf{E_W}}, \widehat{\mathbf{D}}$ denote embedding matrices, attaching coefficients from $X, W$ to group elements and vice-versa. Then $f(X, W) = \mathbf{D}^\top \mathbf{F}^{-1}((\mathbf{FE_X} \cdot \text{vec}(X)) \odot (\mathbf{FE_W} \cdot \text{vec}(W))$. In words, the operator maps $t \times t$ matrices to some $r$-dimensional vectors, on which we perform the group's convolution. Although convolution seems as a very abstract operation, it is closely related to a familiar concept: polynomial multiplication. In fact, convolution in *abelian* groups is just (multi-variate) polynomial multiplication, modulo some monomial.

This view relates to the group-theoretic approach for FMM, which originated in the work of Cohn and Umans (2003). However, their framework restricts the embedding matrices ($\mathbf{E_X}, \mathbf{E_W}, \mathbf{D}$) significantly and does not deal with the task of approximation, while relying on divide-and-conquer to obtain asymptotic speedups. A recent work of Pratt et al. (2025) is, to the best of our knowledge, the first group-theoretic approach for Approximate MATMUL, although Alman and Zhang (2023) also make a step in this approach (formulated differently). The authors present a simple construction of embedding matrices that achieves SoTA tradeoffs between speed and accuracy. Moreover, the authors present a more sophisticated construction that completely beats SoTA tradeoffs against certain common distributions of matrices (like random $\{\pm 1\}$ i.i.d entries).

The significance of Pratt et al. (2025) for our work, is that it lays a theoretical foundation for the capacity of STL to provide good approximation for MATMULs. Moreover, it provides convolution operators which are theoretically good **initialization** points for the optimization process. As this task is extremely non-convex, good initialization is crucial.

All in all, our work steps out of the approximate MM group-theoretic framework presented in Pratt et al. (2025), by freely optimizing over the encoder / decoder matrices, to match the encountered (activation) matrices. To emphasize the last point, note that we are not trying to learn *global* encoder / decoder matrices, but rather *data-dependent* ones.

## 7 DISCUSSION

We believe that that the approach we presented here, together with the preliminary evidence, motivates further research in many directions. First, whether and in what cases, can STL improve both *accuracy* and *inference throughput* of deep networks. Second, how to *train* STL, and in particular, finding clever *initialization* points and suitable *regularization* techniques. Third, if the random Strassen subset approach can be proved theoretically (against any matrix). Fourth, how well can STL perform with specialized CUDA kernels and dedicated engineering.

While we do not claim SoTA results here, we believe this line of work has the potential to reach or surpass competitive baselines with more study, specifically into the optimization problem STL poses.

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

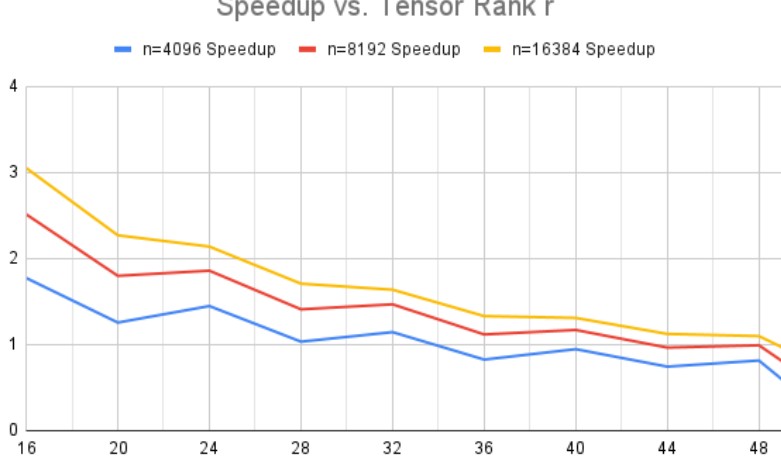

Figure 5: Observed speedup factor for STL for tile size $t = 4$, $r = 16, 20, \ldots, 48, 49$, and matrices of size $n \times n$ with $n = 4096, 8192, 16384$, using a (non-optimized) `PyTorch` implementation. As expected, the throughput speedup is almost linear in the tensor rank. (Note: $r = 49$ can imitate exact matrix multiplication by Strassen, and is given here for completeness.)

## A   GPU COMPLEXITY OF STL

*Proof of Claim 3.4.*

$$Y_{I,J}^{(p)} = \left( \sum_{L=1}^{k/t} (\mathbf{E_X} \cdot \mathsf{vec}(X_{I,L}) \odot (\mathbf{E_W} \cdot \mathsf{vec}(W_{L,J})) \right)_p$$

$$= \sum_{L=1}^{k/t} (\mathbf{E_X} \cdot \mathsf{vec}(X_{I,L})_p \cdot (\mathbf{E_W} \cdot \mathsf{vec}(W_{L,J})_p$$

$$= \sum_{L=1}^{k/t} X_{I,L}^{(p)} W_{L,J}^{(p)} = (X^{(p)} W^{(p)})_{I,J}.$$

$\square$

## B   MORE DETAILS ON THE PARAMETER INCREASE OF STL

In the downstream AI applications of matrix multiplication, we are free to optimize directly in the space of the encoded weight matrix $\widehat{W}$, containing $r$ parameters per tile, instead of $t^2$. This can improve the expressivity of STL as a module inside a network. It turns out that this indeed can be done, as we show in the following sections in the context of training STL inside an actual deep network. However, we first show a negative result. Lemma B.1 below states that, as long as we measure the accuracy of STL using Frobenius norm of the residual (error matrix) with respect to standard matrix multiplication, we effectively do not gain more than $t^2$ trainable weights per tile of $\widehat{W}$, which is the same as the number of parameters of the corresponding tile (the original tile of $W$). The lemma however does not rule out increased expressivity when training using other loss functions, as our experiments in what follows support.

To explain our result, consider the simplified setting of approximating matrix multiplication of two single tile matrices, $X, W \in \mathbb{R}^{t \times t}$. The matrices $X$ can come from any fixed distribution $\mathcal{D}_X$. The matrices $W$ are drawn uniformly from a finite population of size $N$, which we denote $\mathcal{W}$, and the two matrices are drawn independently of each other. To connect this to actual applications, one should think of $\mathcal{W}$ as a collection of tiles from a pretrained weight matrix of some linear layer which we want to replace with the STL operator, which is the STL-equivalent of matrix pruning. The

mathematical reason we restrict $\mathcal{W}$ to be finite is that we want to allow the encoding parameters of $W \in \mathcal{W}$ to be any function, without requiring any structure such as linearity or even smoothness. In other words, the encoding parameters will simply be memorized. The training will optimize over the encoder $\mathbf{E_X}$, the decoder $\mathbf{D}$ and over these fake encoding parameters. Our notation:

$$\text{FakeEnc}(\mathcal{W}) = \{\text{FakeEnc}(W) \in \mathbb{R}^r \mid W \in \mathcal{W}\} . \tag{4}$$

There is now no need for the $W$-encoder $\mathbf{E_W}$. The collection of all values $\text{FakeEnc}(W)$ for $W \in \mathcal{W}$, which be formally denote by $\text{FakeEnc}(\mathcal{W})$, can be thought of, for computational convenience, as a matrix of shape $N \times r$. For a fixed repertory $\mathcal{W}$, the optimization now becomes

$$\alpha_{\mathsf{STL}}^{\mathcal{W}} = \inf_{\substack{\mathbf{E_X}, \mathbf{D} \\ \text{FakeEnc}}} \mathbf{E}_{X,W}[\text{err}(X, W, \mathbf{D}, \mathbf{E_X}, \text{FakeEnc}(W))] \tag{5}$$

where the expectation is over $X \sim \mathcal{D}_X$ and $W$ uniform from $\mathcal{W}$, and the error function $\text{err}(X, W, \mathbf{D}, \mathbf{E_X}, \text{FakeEnc}(W))$ is the mean average error of the residual:

$$\frac{1}{t^2} \|\text{vec}(XW) - (\mathbf{D}^T(\mathbf{E_X}\,\text{vec}(X) \odot \text{FakeEnc}(W)))\|_2^2 \tag{6}$$

The fake encoding variables seem to promise an increase in capacity of the learning space we are trying to optimize over, compared to learning over $\mathbf{E_X}, \mathbf{E_W}, \mathbf{D}$. Unfortunately, as the following lemma reveals, this is not the case, and the reason for this is the choice of the Frobenius norm (squared) loss function. We state and prove this disappointing fact as a lemma, but prepare the disappointed reader that in what follows, the fake encoding parameters will show some promise in downstream AI applications, where the loss functions are different.

**Lemma B.1.** *For any fixed $\mathbf{E_X}, \mathbf{D}$, the optimal value of $\text{FakeEnc}^*(\mathcal{W})$ minimizing the RHS of (5) is given by the relationship $\text{FakeEnc}^*(W) = \mathbf{F}\,\text{vec}(W)$ for all $W \in \mathcal{W}$, for some $\mathbf{F} \in \mathbb{R}^{r \times t^2}$ (which we may as well call the effective encoding matrix for $W$.)*

*Proof.* The first thing to note about (5) is that the optimization problem can be done independently for each $W \in \mathcal{W}$. Hence let us fix one $W \in \mathcal{W}$ and assume that $\mathbf{E_X}, \mathbf{D}$ are such that the minimizer for (5) is achieved. Now define the corresponding minimization problem specific for $W$:

$$\alpha_{\mathsf{STL}}^{\mathcal{W}}(W) = \inf_{\substack{\mathbf{E_X}, \mathbf{D} \\ \text{FakeEnc}(W)}} \mathbb{E}_X[\text{err}(X, W, \mathbf{D}, \mathbf{E_X}, \text{FakeEnc}(W))] \tag{7}$$

Then clearly $\alpha_{\mathsf{STL}}^{\mathcal{W}} = \frac{1}{N} \sum_{W \in \mathcal{W}} \alpha_{\mathsf{STL}}^{\mathcal{W}}(W)$. Now let us replace the vector norm in err by its definition, summing squares over all coordinate differences, so err becomes:

$$\frac{1}{t^2} \sum_{i=1}^{t^2} (\text{vec}(XW)_i - \mathbf{D}^T(\mathbf{E_X}\,\text{vec}(X) \odot (\text{FakeEnc}(W))_i))^2 \tag{8}$$

The expression $\text{vec}(XW)_i$ is clearly a linear function of $\text{vec}\,W$, with coefficient vector we denote by $\mathbf{Z}_{X,i} \in \mathbb{R}^{t^2}$ that depends on $X$ and $i$ only. Similarly, the expression $\mathbf{D}^T(\mathbf{E_X}\,\text{vec}(X) \odot (\text{FakeEnc}(W))_i)$ is a linear function of $\text{FakeEnc}(W) \in \mathbb{R}^r$, with a coefficient vector $\mathbf{Z}'_{X,i}$ that depends on $X, i$ only. (Recall that we assume fixed and optimal encoder $\mathbf{E_X}$ and decoder $\mathbf{D}$ in the premise of the lemma, so we omit them in the notation for $\mathbf{Z}, \mathbf{Z}'$). This allows us to write err as

$$\mathbb{E}_i(\mathbf{Z}_{W,i}^T\,\text{vec}(W) - \mathbf{Z}_{W,i}'^T\text{FakeEnc}(W))^2 \tag{9}$$

where the index $i$ is uniformly taken in $[t^2]$. The optimization now becomes that of minimizing:

$$\mathbb{E}_{X,i}(\mathbf{Z}_{W,i}^T\,\text{vec}(W) - \mathbf{Z}_{W,i}'^T\text{FakeEnc}(W))^2 , \tag{10}$$

over the $r$ variables $\text{FakeEnc}(W)$. Now it is clear that the last minimization is a linear regression with $r$ variables over a distribution of equations. The optimizer $\text{FakeEnc}^*(W)$ is given by

$$\text{FakeEnc}^*(W) = \underbrace{\mathbf{E}_{X,i}\left[\mathbf{Z}'_{X,i}\mathbf{Z}_{X,i}'^T\right]^{-1} \mathbf{E}_{X,i}\left[\mathbf{Z}'_{X,i}\mathbf{Z}_{X,i}^T\right]}_{\text{Solution Matrix}} \text{vec}(W). \tag{11}$$

The Solution Matrix of shape $r \times (t^2)$, independent of $W$, mapping the original matrix $\text{vec}(W)$ to its optimal fake encoding, is effectively the desired encoding matrix $\mathbf{F}$ from the Lemma statement. $\quad\square$

The underwhelming implication of Lemma B.1 is that, when measuring the approximation error of STL vs. MATMUL in the $L^2$ *norm*, one cannot gain expressivity from the use of the extra learnable parameters hidden in the fake encoding of the $W$ matrices, compared to the expressivity we get from using a linear encoding function $\mathbf{E_W}$ to encode $W$. Notice also that the proof did not use the fact that we were working over single tiny $t \times t$ tiles. It just uses the fact that, viewed as a function on activation matrices $X$, the STL operator for fixed $(W, \mathbf{E_X}, \mathbf{E_W}, \mathbf{D})$, is a linear operator. The conclusion from Lemma B.1 would hold true for matrices of any shape, and lead to the conclusion: Directly optimizing fake encoding parameters for the tiles of a weight matrix $W$ does not effectively buy us more parameters than those already present in the original matrix $W$, as long as we care about Frobenius norm of the MATMUL error.

Interestingly, when training STL for LLM downstream tasks, the actual loss function we are working with is the *perplexity of language prediction* (Chen et al., 2018), which is quite different than the (layer-wise) $L^2$ norm (Wenger et al., 2023). Indeed, our experiments involving training LLMs from scratch using STL show the effect of training STL layers in the (fake) encoding space, reassuring that it *does exploit the parameter increase* of the operator.

## C EXPERIMENTS

### C.1 CLASS 0 EXPERIMENT: COMPARING STL TO 2:4 ON RANDOM SYNTHETIC DATA

In our first experiment, we compared the accuracy of STL with tile size $t = 4$ with various parameters on matrices of size $4 \times 4$ (corresponding to a single tile), with different values of $r$, to that of structured 2:4 pruning. The main technical difficulty of this experiment was training the encoder and decoder matrices $\mathbf{E_X}, \mathbf{E_W}, \mathbf{D}$. As we shall see below, a gradient descent learning strategy is highly dependent on the initialization of the solution.

We will concentrate on tile size $t =$ and $4 \times 4$ matrices. To define the loss for the 2:4 benchmark, we define a mask operator $\mathcal{M}$ which identifies the 2 highest (in magnitude) coordinates of each column of $W$, more precisely,

$$\mathcal{M}(W)_{ij} = \begin{cases} 1 & i \in \mathrm{ArgTop2}\{|W_{kj}|\}_{k=1..4} \\ 0 & \text{otherwise} \end{cases} \tag{12}$$

where $\mathrm{ArgTop2}$ returns the two indices of the largest (in absolute value) two elements in a list of elements, breaking ties (say) by preferring lower indices.

The quality of this approach is denoted $\alpha_{2:4}$ and is defined as follows:

$$\alpha_{2:4} := \frac{1}{16} \mathop{\mathbf{E}}_{W} \min_{\widetilde{W} \in \mathbb{R}^{4 \times 4}} \mathop{\mathbf{E}}_{X} \|XW - X(\widetilde{W} \odot \mathcal{M}(W))\|_F^2.$$

The $1/16$ factor gives the average (since we are working with $4 \times 4$ tiles). Moreover, we minimize over $\widetilde{W}$, to allow more advanced 2:4-sparsification techniques, which take the training data into account. Note that if $\mathcal{D}_X$ was just the uniform distribution over all matrices (with bounded norm), then the solution would have always been $\widetilde{W} = W$.

For a fixed $W$ matrix, the minimizer for $\widetilde{W}$ in the last equation can be easily approximated by solving a convex program (in fact, a linear regression problem) over a random large (but fixed) population of $X$'s. Our experiments have resulted in the following estimate:

$$\alpha_{2:4} \approx 0.53.$$

Our goal is to obtain a competitive error for approximation of $XW$ using STL. It should be noted that our approximation is dependent on the distributions $\mathcal{D}_X, \mathcal{W}$. For the sake of our experiment, we set $\mathcal{D}_X$ to be matrices whose entries are i.i.d. from $\mathcal{N}(0, 1)$ (normal Gaussian distribution, mean 0, variance 1). In general, if one wishes to approximate a linear layer in a trained network with STL, it could make sense to take the distribution of $W$ to correspond to the empirical distribution of tiles of the pretrained weight matrix, and that of $X$ to come from the actual data of interest flowing through the network.

We similarly define the quality of the STL approximation to be

$$\alpha_{\mathsf{STL}} = \min_{\mathbf{E_X},, \mathbf{E_W}, \mathbf{D}} \big[ \mathop{\mathbf{E}}_{X, W} \mathrm{err}(X, W, \mathbf{D}, \mathbf{E_X}, \mathbf{E_W}) \big], \tag{13}$$

where err is defined as before, only replacing $\mathrm{FakeEnc}(W)$ with the $W$-encoder (recall that we gain nothing by using fake encoding parameters, by Lemma B.1, at least in the $L^2$ sense);

**Estimates of $\alpha_{\mathsf{STL}}$.** We have estimated $\alpha_{\mathsf{STL}}$ w.r.t. the Gaussian distribution on $X$ and a fixed random, Gaussian distributed population of $W$'s by running gradient descent on the encoders and decoders in an attempt to solve the minimization problem defining $\alpha_{\mathsf{STL}}$. The results were summarized in Figure 1 in the main part of the paper. It turns that out estimates heavily depend on the initialization of the gradient descent algorithm (see below for more details).

It appears from the plot that for approximately $r = 42$, $\alpha_{\mathsf{STL}}$ roughly matches $\alpha_{2:4}$. From Figure 2 in the main part of the paper, it is evident that at $r = 42$ there is little chance to beat $2:4$ in performance. However, the following should be noted: Our estimation of $\alpha_{2:4}$ is very accurate, because it is calculated by averaging out over a random population of weight matrices $W$, an estimation of the $2:4$ pruning error, which is a convex problem.[4] Therefore, our comparisons are $\mathsf{STL}$-optimistic in the sense that it is likely that the true optimal bounds for $\alpha_{\mathsf{STL}}$ are better, possibly using better initialization and/or optimization techniques. This is in fact one of the main open questions in this paper.

**Initialization Issues for Class 0 Experiment.** To estimate $\alpha_{\mathsf{STL}}$, we solved a non-convex optimization problem over the encoders and decoders, using gradient descent. Initializing the encoder and decoder parameters randomly gave us suboptimal estimates, compared to the following method, which is based on a pruned version of Strassen's encoders and decoders used for getting a tensor of rank 49 for multiplying a pair of $4 \times 4$ matrices.

If $\mathbf{E_X}^{49}, \mathbf{E_W}^{49}, \mathbf{D}^{49} \in \mathbb{R}^{49 \times 16}$ denote the encoders and decoders for Strassen's construction, then our construction for initializing the optimization for $\alpha_{\mathsf{STL}}$ was done by simple random *pruning* in the encoding-space dimension. More precisely, we chose a random subset $I$ of $r$ integers in $[49]$ (without repetitions), and initialized $\mathbf{E_X}, \mathbf{E_W}, \mathbf{D} \in \mathbb{R}^{r \times 16}$ to be the matrices obtained by extracting the $r$ rows indexed by $I$ from $\mathbf{E_X}^{49}, \mathbf{E_W}^{49}, \mathbf{D}^{49}$, respectively. This rather naive initialization heuristic already gave significantly better results than random initialization.

### C.2   CLASS 1 EXPERIMENT: TRAINING T2T-VIT WITH STL

**More Details on $\mathsf{STL}$ Replacement in T2T-ViT**   In the ViT architecture, and in particular in T2T-ViT (Yuan et al., 2021b), the input image is organized as patches. In our case each patch is $16 \times 16$ in resolution, resulting in a two dimensional spatial *patch* space of shape $14 \times 14$ for images of original resolution $224 \times 224$. Each patch corresponds to a *token* in the language of transformer networks. In addition to the $14 \times 14 = 196$ tokens, an additional "summary" token is appended and used at the end for classification. This results in 197 tokens representing an instance image in the attention network pipeline.

There are two technical challenges with this token space, when viewed under the $\mathsf{STL}$ lens.

1. $\mathsf{STL}$ with tile size $t = 4$ packs together every $4$ coordinates of the (activation) matrix, and 197 is not divisible by 4. We chose to solve this by appending another $3$ *null* rows to the activation input matrix $X$ (for each $\mathsf{STL}$ layer). When obtaining the output matrix $Y$, we reduce the dimension from 200 back to 197 by linearly combining the last $4$ rows into a single row, using another $4$ trainable coefficient parameters. There are other natural choices for this technical detail. For example we could use $4$ summary tokens instead of one, but our choice seemed to be the simplest.

2. In the original ViT network architecture, the patches are organized in raster order, and therefore each $\mathsf{STL}$ tile packs together $4$ patches that visually correspond to a horizontal slab of length $4$ patches. The choice of horizontal (vs. vertical) seems quite arbitrary, and we felt that it should not affect the inductive bias of the network. Hence we have reorganized the order of patches, so that each $2 \times 2$ *square* of $4$ patches would be contiguous in memory, and hence in the activation matrix indexing. This is done once before the attention pipeline and has negligible IO cost, which will become more negligible for larger ViTs.

---

[4]This is after having chosen the pruned coordinates using the magnitude heuristic. We are aware that there are more advanced methods for pruning, but (a) it is not clear whether those methods really make a difference for $4 \times 4$ matrices and (b) there are possibly more advanced ways to optimize for $\alpha_{\mathsf{STL}}$.

**Preliminary Results in the Over-Parameterized Regime**  As we've stated before, we are not aware of other work that reports improvement on the Imagenet-1K classification problem, when trained from scratch on the Imagenet-1K training split with a network of similar size and considerable matrix pruning (in particular, structure pruning). The closest reported results we are aware of are Chen et al. (2021) which thoroughly studied pruning strategies of a related architecture called DeiT-Vision-Transformer. For a model DeiT-Tiny of a similar size as T2T-ViT-7, all their pruning experiments led to more than $2\%$ degradation of accuracy, even at only $30\%$ unstructured sparsity rate, let alone with $2:4$ (structured) sparsification.

The cases where they saw accuracy gains in from sparsification were on DeiT-Base which has roughly 80M parameters ($\times 4$ parameters compared to T2T-Vit-14). We argue that, for that size model on Imagenet-1k, the over-parameterization is so extreme that sparsification possibly helps by virtue of the regularization it offers. This is also confirmed by a followup experiment that we did on the T2T-ViT-14 architecture (21.5M parameters, 6.1G FLOPS per 224x224 image) from the same paper Yuan et al. (2021b).

For this model we lost between $2\%$ and $3\%$ accuracy when replacing with STL, compared to baseline, for all values of $r$ ranging from 16 to 49. Recall that at $r = 49$ there is provably no loss of expressivity, because STL at that tensor rank allows expressing exact matrix multiplication (by Strassen), and hence the empirical *loss* of accuracy in this case is probably due to the *extreme over-parameterization* owing to the effective increase in parameters.

## D  PSEUDO-CODE FOR STL

For ease of notation, we let $\widehat{X}$ and $\widehat{W}$ denote the encoded versions of $X, W$, i.e., a tensor of size $(n/t, n/t, r)$ with $\widehat{X}[I, J, :] = \mathbf{E_X} \cdot \text{vec}(X_{I,J})$ (the encoding of the $I, J$-th tile). Similarly for $\widehat{W}$. We let $Y = X \diamond W$, and $\widehat{Y}$ denote the encoded tensor.

We note that in `PyTorch`, when trying to multiply the last dimension of a 3D tensor by a matrix, this is done in transposition to the clean mathematical formulation. In other words, to compute $\widehat{X}$, we need to compute $\mathbf{E_X} \cdot X[I, J, :]$ for every $I, J$, where we view $X[I, J, :]$ as a $t^2$ column vector. In `PyTorch` this is done by `hatX = X  E_X.T`, which can be interpreted as viewing $X[I, J, :]$ as a row vector of size $t^2$, and so the product with $\mathbf{E_X^\top}$ gives a new row vector of length $r$. The algorithms are written in a mathematical formulation first, and `PyTorch` formulation second. We also provide a `PyTorch` implementation in our public repository.

---

**Algorithm 1** STL Pseudo-code (GPUs)

---

**Require:**

  Tensor $X$ of shape $(n/t, n/t, t^2)$ (Each tile flattened)

  Tensor $\widehat{W}$ of shape $(n/t, n/t, r)$ (Each tile encoded)

  Encoding matrix $\mathbf{E_X}$ of shape $(r, t^2)$

  Decoding matrix $\mathbf{D}$ of shape $(t^2, r)$

  **Step 1: Encode**

  **for** $I, J \in [n/t]$ (in parallel) **do**

    $\widehat{X}[I, J, :] \leftarrow \mathbf{E_X} \times X[I, J, :]$

  **end for**

  (In `PyTorch`: `hatX = X @ E_X.T`)

  **Step 2: Batched Element-wise Product**

  **for** $p \in [r]$ (in parallel) **do**

    $\widehat{Y}[:, :, p] \leftarrow \widehat{X}[:, :, p] \times \widehat{W}[:, :, p]$

  **end for**

  (In `PyTorch`: `hatY = (hatX.permute(2,0,1) @ hatW.permute(2,0,1)).permute(1,2,0)`)

  **Step 3: Decode**

  **for** $I, J \in [n/t]$ (in parallel) **do**

    $Y[I, J, :] \leftarrow \mathbf{D}^\top \times \widehat{Y}[I, J, :]$

  **end for**

  (In `PyTorch`: `Y = hatY @ D`)

  **Return:** $Y$ (Each tile flattened)

---

---

**Algorithm 2** STL Pseudo code for fused Steps 3+1, at layer $\ell$

---

**Require:**

  Tensor $\widehat{X}_{\ell-1}$ of shape $(n/t, n/t, r)$ (Previous layer encoded output activations)

  Tensor $\widehat{W}_\ell$ of shape $(n/t, n/t, r)$ (Encoded weights for this layer)

  Encoding matrix $\mathbf{E_X}$ of shape $(r, t^2)$

  Decoding matrix $\mathbf{D}$ of shape $(t^2, r)$

  **Steps 3+1:**

  **for** $I, J \in [n/t]$ (in parallel) **do**

    $\widehat{X}_\ell[I, J, :] \leftarrow (\mathbf{E_X} \times \mathbf{D}^\top) \times \widehat{X}_{\ell-1}[I, J, :]$

  **end for**

  (In `PyTorch`: `hatX_this = hatX_prev @ (D @ E_X.T)`)

  **Step 2:**

  **for** $p \in [r]$ (in parallel **do**

    $\widehat{X}_\ell[:, :, p] \leftarrow \widehat{X}_\ell[:, :, p] \times \widehat{W}_\ell[:, :, p]$

  **end for**

  **Return:** $\widehat{X}_\ell$

---

