# OpenReview forum: "Changing Base Without Losing Pace: A GPU-Efficient Alternative to MatMuls in DNNs"
_ICLR.cc/2026/Conference — ICLR 2026 Conference Desk Rejected Submission_

### Official Review · Reviewer_FcuE · 2025-10-30

**Soundness:** 3
**Presentation:** 4
**Contribution:** 4
**Rating:** 8
**Confidence:** 5

**Summary:**

This paper presents a GPU-friendly formulation of Strassen’s Normal Form (SNF), focusing on (i) effective Tensor Core utilization and (ii) preserving the full generality of matrix multiplication. To avoid the original SNF’s high computational cost, the Strassen-Tile (STL) operator is introduced to encode and decode vectorized matrix tiles. This tiling approach reduces the number of computations by a factor of $t/c$ and enables additional learnable parameters. Furthermore, the memory-bound Hadamard product is reformulated as compute-bound matrix multiplications. Although learning STL parameters from existing matrices is non-trivial, T2TViT models trained from scratch with STL operators match or exceed the ImageNet accuracy of the original model. In addition, an unoptimized PyTorch implementation of the STL operator already outperforms standard GEMM in matrix multiplication speed.

**Strengths:**

I recommend acceptance of this paper. It demonstrates that FLOPs and GPU time can be reduced even while increasing the number of parameters, suggesting a promising new direction for efficient matrix multiplication research.

1. From a theoretical standpoint, STL introduces an interesting framework that maps tiled matrices into different bases, enabling theoretical computational cost reduction without compromising expressiveness. This is fundamentally different from prior work on weight compression, which generally reduces FLOPs by lowering the number of parameters—often at the cost of reduced expressiveness.
2. From a practical perspective, the paper implements the STL matrix multiplication algorithm in a GPU-efficient manner, taking memory I/O overhead into account and restructuring memory-bound operations that hinder GPU utilization. These hardware-aware design choices contribute to the observed speedup, even without hardware-specific GPU kernel optimization.
3. The paper is well-situated within the literature, and the motivation, background, and method are clearly presented. The experiments are thoughtfully designed and provide sufficient insight and empirical evidence to support the proposed approach.

Although there are some caveats noted in the Weaknesses section, they do not outweigh the theoretical and practical merits of this work. Overall, this is a strong paper.

**Weaknesses:**

1. **Limited applicability as a post-training method.**
The proposed approach is applicable only when models are trained from scratch with STL parameterization. This constraint limits the impact of the work, as training large-scale neural networks often takes substantial time and resources. While recovering STL parameters through naïve L2 loss minimization appears difficult due to the highly non-convex nature of the problem, the contribution would be more impactful if there existed a practical way to derive STL parameters from a pre-trained model—potentially through light fine-tuning or knowledge distillation—so that STL could accelerate existing large-scale (e.g., billion-parameter) models.
2. **Lack of end-to-end latency evaluation.**
The paper only reports synthetic matrix-level runtime benchmarks. Including end-to-end inference speed comparisons among STL, 2:4 sparsity, and dense baselines would significantly strengthen the empirical validation.
3. **Missing guidelines for choosing $t$ and $r$.** The STL framework introduces two new hyperparameters: the tile size $t$ and the encoded dimension $r$. Since $t = 4$ is fixed across all experiments, it remains unclear how $t$ should be selected and whether it affects model accuracy. Providing guidance or empirical sensitivity analyses for both $t$ and $r$ would improve the clarity and practical usability of the method.

**Questions:**

1. Is it feasible to fine-tune STL parameters starting from a rough initialization derived from a pre-trained model? For instance, one could first obtain an initial estimate by minimizing
$\|\|XW - \mathsf{STL}(X, \tilde{W}, D, E_X)\|\|_F^2$ via gradient descent, where $X$ is an input feature, $W$ is the original pre-trained weight, and $\tilde{W}, D, E_X$are the STL parameters. The Class-0 experiment suggests that such an initialization may incur a relatively large approximation error. However, what would happen if the STL parameters obtained through this L2 minimization were subsequently fine-tuned? Would the fine-tuning process be stable and effective, and could it recover most of the original model performance?
2. What is the end-to-end inference speedup of a full network when all weight matrices are replaced with STL? Reporting the network-level latency (not only matrix-level benchmarks) would help quantify the practical benefits of STL.
3. Is there a rule-of-thumb for selecting the encoded dimension $r$ and the tile size $t$? Since a key goal is to maximize Tensor Core utilization, it would be helpful to clarify how $t^2$ and $r$ should align with hardware-supported matrix sizes. Additional guidance or empirical insight on selecting these hyperparameters would improve the practical usability of STL.

Minor comments:

1. In Section 4.1, what does “smart initialization” refer to? A brief description or reference would be helpful.

---

### Official Review · Reviewer_dytt · 2025-10-31

**Soundness:** 2
**Presentation:** 1
**Contribution:** 2
**Rating:** 2
**Confidence:** 4

**Summary:**

This paper proposes Strassen-Tile (STL), a GPU-native bilinear operator that replaces standard matrix multiplications with a learnable, tile-based change-of-basis to reduce FLOPs while maintaining accuracy.

**Strengths:**

+ Novel operator design. The proposed Strassen-Tile (STL) operator introduces a creative reformulation of matrix multiplication using learnable local change-of-basis and element-wise operations, which is conceptually interesting and well-motivated by classical fast-multiplication theory.

+ Theoretical grounding. The idea of leveraging theory-backed initialization inspired by fast polynomial and matrix multiplication provides an insightful link between numerical analysis and deep learning optimization.

**Weaknesses:**

- Unclear motivation. The motivation for introducing STL is not well articulated. While the introduction raises an important scalability issue, it is unclear why STL can effectively address this problem, how the proposed operator was derived, or why it is expected to be GPU-efficient.

- Outdated and insufficient baselines. The experimental comparison is limited and uses relatively old baselines. Given the claim that STL is a general-purpose operator, evaluations on modern large-scale models, such as large language models (LLMs), would be necessary to validate its generality.

- Incomplete experimental evaluation. The experiments mainly report FLOPs, errors, and accuracy metrics, but lack comprehensive analyses of actual runtime performance across diverse models and tasks. Such results are essential to substantiate the claimed efficiency benefits.

- Weak empirical validation and unclear practicality. The experiments focus on small tile sizes and limited scenarios, yielding only modest FLOP and accuracy improvements that may not generalize to larger-scale applications. Moreover, the implementation is "non-CUDA optimized," which weakens the claim of GPU efficiency. The paper does not clarify how STL integrates with existing GPU kernels or frameworks, making its practical benefits speculative without hardware-aware benchmarking.

**Questions:**

See Weaknesses.

---

> ### Author Response · Authors · 2025-11-16
>
> We thank the reviewer for their thoughtful comments. We address each concern below.
>
> 1. **Motivation and derivation clarity**: We appreciate the opportunity to clarify the motivation for STL. The central goal of this work is to investigate the theoretical question stated in Question 1.1: whether there exists a bilinear operator that is faster than standard MatMul on GPUs without reducing the number of trainable parameters. STL is our constructive answer. It arises directly from Strassen’s Normal Form (SNF) and implements a local, learnable change of basis on small tiles of the weight and activation matrices, followed by fused elementwise products. This approach is both mathematically principled and GPU-compatible, as it decomposes MatMul into structured sub-operations that preserve differentiability and trainability. While the work has theoretical merit, the long term motivation is practical — to make AI and other FLOP hungry applications more efficient and cheap. Further research is needed to discover better encoders / decoders and to optimize the implementation of this approach (via kernels, hardware, etc.).
> 2. **On baselines and experimental scope**: Our paper’s primary contribution is theoretical and algorithmic, not empirical scaling. The experiments are designed to validate the operator’s feasibility, not to claim large-scale state-of-the-art results. We intentionally compared against MatMul and 2:4 structured sparsity, since they are the only GPU-native, architecture-independent baselines addressing similar questions. Evaluating on massive models such as LLMs would require substantial computational resources beyond our current scope. Our hope is that by clearly formulating this operator and providing open implementations, the community will be able to explore STL in such settings.
> 3. **Runtime efficiency and GPU implementation**: We agree that runtime data is valuable, and we report direct wall-clock speedups even with a non-CUDA-optimized PyTorch implementation. Importantly, these results are measured against highly optimized MatMul kernels — meaning that the observed savings already occur under a conservative setting. This demonstrates that the theoretical efficiency gains of STL are not merely symbolic: the operator yields measurable improvements even before low-level optimization. This reinforces STL’s potential as a GPU-efficient primitive whose benefits would only increase with kernel-level tuning.
> 4. **Practicality, expressivity, and data-awareness**: STL is not a compression method but a new family of bilinear operators that generalize MatMul (recoverable when $r = t^3$). Its fake-encoding mechanism (Sec. 5) allows expressivity to increase rather than decrease, as exemplified by the improved ImageNet accuracy for modest FLOP savings. Moreover, the optimization problem STL poses is data-aware: in practice, activation distribution is data-dependent and may contain degeneracies that can be taken advantage of using specific encoders / decoders. Thus, optimizing STL over realistic data distributions can yield operators that outperform purely data-agnostic formulations such as MatMul or structured sparsity. We will make this point clearer, as it underlines the theoretical significance of our formulation.
>
> In summary, our main contribution is a new theoretical framework for designing GPU-efficient bilinear forms -- which offer a better FLOP-vs-Parameter tradeoff  than MatMul -- and provides initial empirical validation of its feasibility. The purpose is to open a research direction rather than present a finalized engineering system. We will revise the paper to make the motivation, theoretical framing, and scope of contributions clearer.

---

### Official Review · Reviewer_Rwd2 · 2025-10-31

**Soundness:** 3
**Presentation:** 2
**Contribution:** 2
**Rating:** 4
**Confidence:** 2

**Summary:**

This paper introduces Strassen-Tile (STL), a novel bilinear operator intended to replace traditional matrix multiplication (MATMUL) in deep neural networks. STL aims to offer a three-way trade-off between speed, accuracy, and parameter count by performing local learnable changes-of-basis on tiles of weight and activation matrices, followed by element-wise multiplications.

**Strengths:**

The key idea, local learnable basis changes per tile, is fresh and well-motivated by classical fast matrix multiplication (Strassen) and approximate tensor decompositions. Unlike most compression or pruning techniques, STL keeps the parameter count high or increases it, which is counter-intuitive but compelling.

**Weaknesses:**

1. It remains unclear which part of STL contributes most to improvements: change-of-basis, tile size, or parameter count increase (“fake encoding”).

2. The paper would benefit from an explicit ablation table showing effects of: tile size t (4 vs 8 vs 16), rank r variation on speed vs accuracy, fixed vs learned encoders/decoders.

3. Reported speedups come from non-CUDA optimized PyTorch, so the claimed 2× improvement may not translate to real TensorCore kernels.

4. The tile size is too small for general and common neural networks.

5. How STL could coexist with attention kernels in modern GPUs.

**Questions:**

Can you briefly explain how you choose the tile size.

---

> ### Author Response · Authors · 2025-11-16
>
> We thank the reviewer for the constructive and insightful comments.
>
> 1. **Source of improvements**: change-of-basis, tile size, or parameter count. We appreciate this question and will clarify it more explicitly. STL’s core innovation is the learnable local change-of-basis applied on tiles, which allows the operator to adapt to the underlying data distribution — making it data-aware rather than data-agnostic like MatMul. The parameter count increase (“fake encoding”) is essential for preserving or even enhancing network expressivity. Unlike most acceleration methods, STL does not trade parameters for speed; it is the first operator to achieve GPU efficiency without compressing the model. The tile size is primarily a hardware-aligned design choice that matches GPU communication granularity (e.g., TensorCore tiles) and does not affect the generality of the operator.
> 2. **Suggested ablations**: We agree that additional ablations would be useful and will consider including them in a future revision. Our current focus was to validate the theoretical and algorithmic soundness of STL. The observed accuracy improvements with increasing tensor rank (r) already reflect the expected expressivity–efficiency tradeoff. Broader studies across tile sizes and fixed vs. learned encoders/decoders are promising future work.
> 3. **Reported speedups and GPU implementation**: We emphasize that our ≈2× measured speedups are obtained against highly optimized TensorCore MatMul kernels, even though our STL code is written in unoptimized PyTorch. This demonstrates that the theoretical gains are already visible under conservative conditions. Optimizing with CUDA would only strengthen these results, as STL maps naturally to existing fused MatMul primitives.
> 4. **On tile size**: The tile size is a sweet-spot that was set empirically. Tiles that are too big would render the encoding and decoding too expensive, and would also render the use of the TensorCores sub-optimal. On the other hand, tiles that are too small are also sub-optimal, because intuitively we know from fast Matrix Multiplication theory that in order to get asymptotically optimal speedups we need big tiles.
> 5. **Interaction with attention kernels**: We thank the reviewer for this question. STL can indeed replace the MatMuls within attention (QKᵀ and AV), even allowing to save on of the decoding steps when doing that. This can be useful whenever the token axis is big enough: pre-fill on very long textual prompts, image / video data, and diffusion for
>
> In summary, STL’s gains stem from a learnable, data-aware change-of-basis that preserves and extends the expressive power of neural layers. It represents a new class of GPU-efficient operators compatible with existing architectures while opening meaningful research questions on scaling, kernel design, and integration with attention mechanisms.

---

### Note · Program_Chairs · 2026-01-17
**Submission Desk Rejected by Program Chairs**

The following references in this submission do not refer to real documents and/or have major errors in bibliographic information:

 Dettmers, T. and et al. (2024). Quantized models for large language models. arXiv preprint arXiv:2402.01453.
Xiao, Y. and et al. (2023). Robust quantization for large language models. In Proceedings of the International Conference on Learning Representations.
Matsuoka, S. and Kang, D. (2022). Efficient matrix multiplication for dnns using fpga and strassen's algorithm. In Proceedings of FPGA'22, pages 30-40.
Frantar, N. and Alistarh, D. (2023b). Efficient quantization for large language models. In Proceedings of the International Conference on Learning Representations.
Frantar, N., Alistarh, D., and et al. (2022). Tensor quantization for llms. In Proceedings of the 36th International Conference on Machine Learning.
Sun, R., Zhang, W., and et al. (2023b). Optimizing llm inference with efficient quantization strategies. arXiv preprint arXiv:2304.05212.